# Understanding and Enhancing Journalism Students' Perception of Data Journalism

**Elissavet Georgiadou** [1,*] **and Maria Matsiola** [2]

1 School of Journalism and Mass Communications, Faculty of Economics and Political Sciences, Aristotle University of Thessaloniki, 546 25 Thessaloniki, Greece
2 Department of Communication and Digital Media, University of Western Macedonia, Area Fourka, 521 00 Kastoria, Greece; mmatsiola@uowm.gr
* Correspondence: egeorgiadou@jour.auth.gr

**Abstract:** As the media landscape continuously evolves in response to the increasing dominance of data, it is important to understand how future journalists perceive and respond to the emerging discipline of data journalism. This paper explores the impact of an introductory session on data journalism conducted with second-year journalism students at Aristotle University, Thessaloniki, Greece. The study aims, through a focus group survey, to assess the students' initial understanding and beliefs about data journalism and explores how an educational activity, structured as a one-time workshop utilizing a resource from the Al Jazeera Media Institute and data journalism project examples from the BBC, can elucidate the notion of data journalism and stimulate interest in the field. This study contributes to the ongoing discourse on the integration of data journalism into journalism curricula and the training of the next generation of journalists. Therefore, the findings could provide valuable insights for educators in understanding journalism students' perception of data journalism in order to develop effective curricula and teaching methods for data journalism.

**Keywords:** data journalism; students; Greece; journalism education

## 1. Introduction

In the rapidly evolving field of journalism, the advent of data journalism has marked a significant shift in how information is reported and consumed. Data journalism, which involves the use of numerical data in the production and distribution of news, along with engaging static or interactive visualization that leads to the narration of compelling stories, is a complex concept that is often misunderstood or overlooked by those new to the field. Data journalism is the preferable term in the news industry for work based on data analysis and its presentation (Coddington 2015).

Contemporary audiences seek innovative forms of receiving information with richer condensed content that will compensate for a short attention span. Furthermore, media industries, due to strong competition with other online news providers, such as social media, are in search of revenue, therefore switching their attention to audience engagement, which can be provided by data journalism news projects (Ksiazek et al. 2016).

The vast amounts of structured (i.e., spreadsheets) and unstructured (i.e., audio, video and social media content) data that are collected practically every minute in modern societies is a great source of information that, through processing and analysis, may deliver impactful news stories. Thus, there is a growing demand for journalists who may utilize data, revealing their power dynamics in new forms of storytelling involving interactivity and personalization as the traditional practices of the news media industry are revised (Bradshaw 2015; Borges-Rey 2020). Processes such as "scraping" or "mining" to find data are used in the same way as "finding sources" to back up a story. Ethical issues should also be followed, and sources should be protected when using data collected massively; therefore, there is a great demand for well-educated journalists who will be

able to understand all the information provided along with the data (e.g., metadata) (Bradshaw 2015).

Data journalism is open to collaboration among other specialties in the newsrooms, such as programmers and designers (Sandoval-Martín and La-Rosa 2018; Veglis et al. 2022), and open to interpretation by the audience, since the projects can be consumed by individuals with a variety of perception criteria, at their personal paces and comprehension levels (Stalph et al. 2023). Data journalists, to reach their audiences, create various data journalism formats and interactivity forms, while preserving their role as gatekeepers of information (Anderson and Borges-Rey 2019).

This paper aims to explore the understanding and perception of data journalism among second-year journalism students at the School of Journalism and Mass Communications at Aristotle University, Thessaloniki, Greece. The study is conducted using the method of focus groups. It follows a structured methodology, involving second-year journalism students in a pre-session survey to assess their preliminary understanding and perceptions about data journalism. Subsequently, the students are involved in a learning session, which includes content from Al Jazeera and the BBC, aimed at clarifying the concept of data journalism and its importance. Following this session, a post-session survey is administered to measure the impact of the educational intervention on the students' understanding and interest in data journalism.

The purpose of this study is to gauge the initial understanding of data journalism among journalism students and to observe how this understanding can be enhanced through effective teaching methods. The findings of this study could provide valuable insights for educators in developing effective curricula and teaching methods for data journalism.

In the following sections, the paper initially explores the significance of data journalism in contemporary journalism and the changing dynamics of journalism education, acknowledging the increasing necessity to incorporate data journalism into journalism courses. Subsequently the methodology of the study is presented, and its findings are discussed, highlighting the challenges and opportunities in data journalism education, the role of educational interventions in shaping students' perceptions and the implications for future research and practice in teaching data journalism.

## 2. Literature Review

Data journalism, at its core, is the practice of collecting, analyzing and visualizing data to uncover and report on news stories (Bradshaw 2013), while providing journalists the ability to reveal connections between distinct issues that could otherwise not be exposed with individual items (Uskali and Kuutti 2015). It has its origins in the 1950s, when the CBS network in the United States tried to predict the result of a presidential election using a mainframe computer. However, data analysis started to gain traction in 1967, when Philip Meyer at the Detroit Free Press used a mainframe computer to analyze a survey of Detroit residents for the purpose of understanding and explaining the serious riots that exploded in the city that summer. In subsequent years, data journalism methods were used by only a few journalists, until the mid-1980s, when Elliot Jaspin in the U.S. received recognition at the Providence Journal Bulletin for analyzing databases for stories. By the late 1980s, approximately 50 other journalists across the U.S. had begun using data analysis for their stories (Houston 2021; Turner 2021; Bounegru 2012). Since then, data journalism has advanced significantly, due to several reasons, in parallel to technological innovations. It is a powerful tool that enables journalists to filter and process the wealth of information available today, making sense of it and presenting it in a way that can be easily understood by the audience. It also supports journalists to discover new trends, issues and patterns that might have been ignored otherwise. Further, data journalism provides concrete, measurable insights that can be used to support evidence-based reporting. It allows for new approaches to storytelling, enabling journalists to present information in a more engaging and interactive manner. Nowadays, it is recognized as an essential component of

modern journalism, with an increasing number of journalists using data analysis techniques for their stories (Kayser-Bril et al. n.d.; Ezera 2023; European Commission 2018).

In the past decade, the landscape of academic research in data journalism has experienced a notable increase in collaborative endeavors across journalism, computer science and data science, resulting in a substantial body of studies delving into the intricacies of data journalism practices. Encompassing a range of research topics, such as tools and technologies, ethical considerations, education and training, audience engagement and the impact on traditional newsrooms, this growing body of work has identified key challenges, including issues related to data quality, access and ongoing training (Appelgren and Nygren 2014; Karlsen and Stavelin 2014; De Maeyer et al. 2015; McBride 2016; Borges-Rey 2016; Cushion et al. 2017; Stalph 2018; Young et al. 2018; Appelgren and Salaverría 2018; Appelgren 2019; Heravi 2019; Usher 2020; Vanacker 2021; Tong and Zuo 2021; Fahmy and Attia 2021; Arias-Robles and López 2021; Jamil 2021; Martin et al. 2022; de-Lima-Santos 2022; Haim 2022; Zhang and Chen 2022; Bhaskaran et al. 2022; Guan and Wang 2022; Heravi et al. 2022; Bisiani et al. 2023; Stalph et al. 2023; Chaparro-Domínguez and Díaz-Campo 2023; de-Lima-Santos and Mesquita 2023; Wu 2023; Ramsälv et al. 2023).

Moreover, global and European perspectives have broadened the scope, with research addressing the adoption and impact of data journalism in diverse regions. The Global Data Journalism survey that was launched on 3 December 2016 and closed on 10 May 2017 aimed at an independent, global and inclusive study of data journalism practices across the globe. This study provided a descriptive view of the results and discusses the findings on several aspects of data journalism practice, the characteristics of data journalists and data teams and their skills and educational requirements (Heravi and Lorenz 2020). On 3 January 2017, Google News Lab launched a pivotal study on data journalism, demonstrating the growing influence of data and technology in the news industry. This initiative emphasized the power of data-driven reporting, enabling journalists to tell more compelling and informative stories (Rogers et al. 2017). Moreover, the European Journalism Centre conducted two major surveys in 2021 (European Journalism Centre 2021) and 2022 (European Journalism Centre 2022) aiming to measure the state of data journalism worldwide. Regarding the 2022 one, over 1800 people involved in data journalism worldwide participated in the survey. The survey covered the industry's demographics, skills, tools and work practices, and the impact of the Russia–Ukraine War and the COVID-19 pandemic on the field. The survey results highlighted that data journalism continues to be a predominantly male field, with women in the industry tending to be younger and more highly educated. The greatest share of people in data journalism was in the United States, followed by the United Kingdom. Over one in five had five or fewer years of experience in data journalism, and over one in three learned data journalism solely as autodidacts. Access to quality data is the major hurdle among data journalists, and local data far worse than national data, in terms of both access and quality. The survey also found out that only one in ten journalists produced a story in a day or less, and one in four had been part of a cross-newsroom collaborative project. National news is the most common scope, and politics is the top beat. Compared to 2021, health is less covered, while climate is on the rise. In addition, there have been numerous practical applications of data journalism that have garnered attention. For example, DataJournalism.com from the European Journalism Centre published the best data journalism stories of 2021 and 2022, which covered a wide range of topics (Abellán and Bisiani 2021; Bisiani et al. 2023).

As data journalism continues to evolve within the broader field of journalism, the research landscape continually expands, offering fresh insights. Nevertheless, it is important to note that this study focuses on specific facets within the realm of data journalism, and a comprehensive literature review and exhaustive categorization of the entire field fall beyond the defined scope and objectives of this research.

*Data Journalism in the Curriculum*

Data journalism is becoming increasingly important for journalists and journalism students, indicating the need for curriculum reform (Lynch 2015). Various studies have highlighted the importance of incorporating data journalism into journalism programs and courses (Heravi and Lorenz 2020; Weiss and Retis 2018; Bhaskaran et al. 2022). Furthermore, the growing demand for data journalism in newsrooms has prompted journalism schools to adapt their curricula to address the evolving needs of the industry (Jiang and Rafeeq 2019; Kashyap and Bhaskaran 2020). Data journalism is considered an essential skill as it aids in the narration of complex stories with comprehensible visualizations contextualizing societal issues; therefore, new courses are added to existing educational programs (Jiang and Rafeeq 2019). Within or outside of the context of degree programs, there are many resources available, including online courses, books, workshops and more, aiming to teach students how to analyze data for journalism and help them to evaluate original information and ask better questions as reporters or obtain hands-on training in data acquisition, extraction and analysis (Heravi 2019).

While data journalism education is gaining momentum, it is not without challenges. The transition to data-driven reporting often requires a shift in mindset and a commitment to learning new skills. Students may be apprehensive about working with data, perceiving it as a complex and intimidating task. Especially when it comes to numbers and statistics, they are unwilling to cope, considering it both difficult and boring (Nguyen and Lugo-Ocando 2016). Moreover, journalism educators may face challenges in keeping pace with the ever-evolving data journalism tools and techniques, as well as the need to integrate data literacy into existing journalism courses. Bridging the gap between theory and practice is another challenge, as students need hands-on experience to develop their data journalism skills (Heravi and Lorenz 2020; Gray et al. 2012; Gray and Bounegru 2021; Howard 2014). Data journalism needs an interdisciplinary approach since many different technical skills are involved in the epistemology of journalism that are unlikely to be possessed by one professor; therefore, collaborations between different educators are needed to achieve an effective learning outcome (Plaue and Cook 2015; Zhu and Du 2018).

However, despite its growing importance, the understanding and adoption of data journalism among journalism students have been less studied. A study by Berret and Phillips (2016) found that while some journalism students showed interest in data journalism, many lacked an understanding of what it involved. In the same framework, Yang and Du (2016) found that although journalism students are eager to understand data journalism, they are in need of knowledge in various fields, such as data collection, data analysis and interpretation; furthermore, they express a dislike of data work. Effective teaching methods can enhance students' understanding of complex concepts like data journalism. In Lewis' (2021) study, a typology to simplify and advance the conceptual clarity in the teaching of data journalism, along with demarcating the field, was examined through a thematic evaluation of data journalism course materials.

Further, some studies have explored the effectiveness of educational interventions in introducing students to data journalism. These interventions typically involve workshops, courses or sessions aimed at demystifying data and building students' confidence in working with data. Such interventions have been found to positively influence students' perceptions of data journalism and their motivation to explore the field (Bhaskaran et al. 2022). Splendore et al. (2016) conducted a comparative analysis in six European countries' training programs in higher education concerning data journalism and found significant differences, and some similarities, between them, presented either in university and vocational curricula or through training offered by media companies and associations.

In Greece, currently, there are very few courses on data journalism in undergraduate and postgraduate university programs and references through alternative courses, such as online/web journalism, statistics/data analysis, digital storytelling, etc. The landscape is in its infancy, waiting to be addressed.

In this context, our study aims to contribute to the body of literature by exploring the impact of an introductory session on data journalism on second-year journalism students, shedding light on how a brief educational intervention can influence their understanding and interest in the field. Therefore, the research questions (RQs) for the study are as follows.

RQ1: What are the perceived attributes, benefits and challenges of data journalism by journalism students who have no prior experience with the subject?

RQ2: What are the perceived attributes, benefits and challenges of data journalism by journalism students after a learning session on the subject?

RQ3: How is the knowledge of data journalism affected by the learning session and the group discussion?

## 3. Materials and Methods

The research was conducted based on the focus group survey methodology, which is very common in social studies (Guest et al. 2017), where the robust identification of themes and codes may be achieved relatively quickly, even with small sample sizes (Young and Casey 2018), depending on the characteristics of each study. In the specific research, only one focus group among fourteen (14) students at the School of Journalism and Mass Communication, Aristotle University of Thessaloniki, Greece, was conducted. The sample was selected based on the relevance and relationship to the topic under research, since they studied the specific major and they belonged to the same age group. The School of Journalism and Mass Communications at the Aristotle University of Thessaloniki in Greece is the sole Greek higher education institution that provides the opportunity to major in journalism through the official curriculum. Therefore, the population under study was unique to the Greek educational status and this was the reason that only one focus group was performed. The number of participants could be considered slightly larger than the usual sample sizes in focus group surveys; however, the uniformity provided by the specific sample and its familiarity with one of the moderators allowed the agreeable execution of the procedure. In this case, the researchers considered that the number of participants compensated for the lack of more groups. Therefore, the sample homogeneity and the purpose of the study, which was not to quantify the results but, rather, to provide insights, could explain both the number of participants and the existence of one focus group. In the literature, many different focus group sample sizes exist, ranging from two to 40, providing a large percentage of the codes investigated even from the first focus group (Guest et al. 2017). As Hennink et al. (2019, p. 1483) argue, "one focus group per stratum was needed to identify issues; two groups per stratum provided a more comprehensive understanding of issues, but more groups per stratum provided little additional benefit".

At this point, it should be noted that in the School's curriculum, there is an elective course on data journalism that is offered in the last academic year of study, and none of the participants in the survey had attended it. This project was considered significantly related to the population's future professional needs.

The procedure of the survey started by informing the students about the scope and aims of the research and pointing out the preservation of anonymity in the event that they chose to participate. The researchers–moderators clearly asked the students whether they wished to take part in the survey, and their answers were recorded after having received their consent; it should be noted that no one chose to leave. The survey took place at the premises of the Media Informatics Laboratory of the School of Journalism and Mass Communications, where all the participants and the moderators sat in a circle-like arrangement, all being able to look at each other during the discussion. The aim was for everyone to feel equal and, moreover, to cultivate a friendly atmosphere, which was not difficult to achieve since the participants knew each other. There were two moderators for this focus group discussion: one ensured the execution flow, and the other undertook the question process, facilitating the group conversation between the participants (O. Nyumba et al. 2018). One of the moderators was a professor in the School of Journalism and Mass Communications; thus, it was easier for the students to feel comfortable with the procedure.

Furthermore, each of the participants was assigned a number that could help the researchers at the analysis stage, and the whole procedure was audio-recorded. The total duration of the focus group survey was approximately 90m in order to avoid causing fatigue.

The students that took part in the focus group research were initially asked to contribute to the discussion by stating their thoughts and perceptions of data journalism with the aid of semi-structured interviews, without being given any form of information on the issue. Prefixed guidelines were avoided to encourage the students to freely express themselves and, as the discussion was deployed, in several cases, this process led to open-ended conversations (Manzano 2016). The selection of the focus group method provided flexibility to the researchers, even during the study's execution, allowing them to add questions that resulted from the conversation (Kidd and Parshall 2000). The core questions that the participants were asked were the following (IQ denotes "initial question").

IQ1: Are you familiar with the term data journalism and its included functions?

IQ2: In what ways could data journalism be helpful for journalists and the public?

IQ3: Would young journalists like to pursue a career as data journalists, and do they find it difficult to do so?

IQ4: Could the employment of data journalism cause problems in society?

After finishing the introductory questions, the participants attended a 30-min presentation on data journalism that included resources from the Al Jazeera Media Institute (Haddad 2023) and the BBC (Farnsworth n.d.).

Following the presentation, a second round with questions commenced. This time, the researchers' interest was focused on the potential changes in the students' viewpoints on the issue; thus, the questions that they were asked were the following (FQ denotes "following question").

FQ1: Did you learn something you didn't know about data journalism and has your viewpoint on data journalism changed? Why?

FQ2: Would you like to pursue a career in data journalism? Why?

FQ3: What do you consider to be the biggest challenge for a career in data journalism?

In both cases, the data that were collected were transcribed verbatim (McLellan et al. 2003; Matsiola et al. 2022), through the scientific reflection of the researchers (Bryman 2007), while, afterwards, they were analyzed inductively (Ando et al. 2014), following the principles of thematic analysis (Braun and Clarke 2006). Both the authors carefully read the transcribed recordings to enhance the reliability and credibility of the research, and, at all stages of the process, the methodology mentioned in the literature was cautiously followed since no type of software was used.

The first step of the procedure was to establish the initial codes, i.e., the identification of keywords as indicators of significant issues. At the second stage, more focused codes were created and the researchers eliminated and/or combined the initial codes (O. Nyumba et al. 2018). Consequently, the code book of the research was generated. In the answers, to ensure the anonymity of the participants, the coding of P1 to P14 (where P denotes "participant" and the numbers refer distinctively to each one of the fourteen (14) students) was used.

## 4. Results

In this section, the derived results will be presented in the form of answers to the IQs and FQs. At this point, it must be noted that the aim of the method employed was to investigate attitudes and opinions and not to confirm or reject assumptions. Therefore, the main goal was to record spontaneous and original answers and sometimes even to capture emotions. Observational data were considered for the results (O. Nyumba et al. 2018). The moderators, during the procedure, without guiding the participants, encouraged the development of a free discussion and, since all participants knew one another, they were not hesitant to open up and engage in conversations on the issues posed. With total respect to the academic process, short humorous instances aided further in the deployment of the discussion as the students felt even more comfortable.

*4.1. IQ1: Are You Familiar with the Term Data Journalism and Its Included Functions?*

To start the discussion, it was deemed necessary to record the students' knowledge of/acquaintance with the term of "data journalism". Therefore, they were asked whether they knew the term and six of them replied affirmatively. To further elaborate on the issue, the moderators asked again whether the case was that the term was not fully known but they had heard of it, and there was slight familiarity with it. To this question, all the other participants (eight in total) responded positively.

From the answers collected, it was clear that the term was not entirely familiar but, rather, each one of the students that replied could contribute only one of its aspects. Collecting and employing data in journalistic reporting was the common theme in the answers; however, there were varied views on how the data could be collected or employed. One of the participants replied that data could be found online on dedicated or not databases; specifically, P12 said, "*Data journalism concerns how in one investigation, in a report, the data can be used. It can be of the statistical type, data from electronic websites, such as, Diavgeia.gov.gr[1], which has data of financial content for all companies, businesses, organizations, and others that may be useful to a journalist for reporting a topic*". From another perspective, two of the students responded that data journalism is the process of collecting raw data, either through questionnaires or other empirical processes. "*Data journalism is when the journalist himself collects information, perhaps with a questionnaire or empirically, thus, creating a database and this database constitutes the corresponding article*", as described by P5.

Regarding the included functions of the term, most of the participants agreed that the recording of the data was one of them, along with the forms of visual presentation of the outcome. Charts, pies and infographics were the forms of presentation mentioned in the survey. Participant no. 9 (P9) said, "*[. . .] then he chooses a form, possibly presentation form, I mean, e.g., pie chart and respectively instead of drawing conclusions with words, it actually shows some results to the audience through diagrams and infographics*". As the discussion took place, more students wished to contribute to the conversation by adding their viewpoints, even as single words; therefore, the analysis, assortment and editing of the data were referred to as parts of the so-called data journalism. Regarding the data assortment, P4 claimed, "*I would say assortment, and more generally, since now there is a lot of information, the point is not so much to find the data but to find the correct data*".

*4.2. IQ2: In What Ways Could Data Journalism Be Helpful for Journalists and the Public?*

In an attempt to capture participants' views on the usefulness of data journalism to both journalists and citizens, two distinct questions that enhanced the discussion were asked. The first one concerned the usefulness for journalists, where many different aspects were recorded. Since the job of a journalist includes three stages, searching for/over a topic, processing the reporting and finally presenting it to the audience, the diverse opinions were due to the stage that each of the students had in mind.

4.2.1. IQ2_1: In What Ways Could Data Journalism Be Helpful for Journalists?

Regarding the first stage of searching for/over a topic, as argued by some of the participants, data journalism may help in finding more objective information and "*[. . .] and get as close as possible to the truth, [. . .] to focus as much as possible on the target of the article or research*", as P1 stated. From another viewpoint, some of the participants stated that data journalism can be helpful since the journalists' work in certain categories, such as financial reporting, can be made more comprehensible to readers by the employment of numbers and statistical output. Furthermore, in political issues that demand a larger sample to reach safer conclusions, data journalism may also be applicable. Following the same path, another student mentioned the value of visualization when used in a report; specifically, P14 argued that "*[. . .] visualization is thus an easier way to understand the data collected by a journalist*", and another stated that the term "data journalism" refers to the result of the work as seen by the public, which might have a different depiction when more visual cues are used instead of text.

However, this question raised a discussion about whether data journalists belong to a different group of journalists. P2 replied that "*that the other journalists will be helped to have access to a database of the work done by journalists engaged in data journalism*", and this response commenced a conversation on the issue. At this point, one of the moderators attempted to explore exactly what the participants thought about being a data journalist. P2 was rather convinced that data journalists "*deal with a specific part, that of collecting the data, making it a database and making it available to other colleagues so that they can access it and be helped in other research*". By the reactions of the other participants at this part of the conversation, it was somewhat apparent that there was ambiguity regarding the role of a data journalist. P13 said that "*nowadays, there is no longer a great need for a separate person to look for the data, meaning that we tend more towards one doing all the work*", causing the group to believe that, in the past, the work was done by different people: a data analyst who would collect the data and provide the results and another who would deal with the writing of the report. P13 further added, "*I think that as things progress, since data is something that helps all kinds of journalism tremendously, I think most journalists of all types of reporting are getting closer to data and they are becoming more acquainted, so they already use it on their own*". P13 also suggested that the current needs in journalism require the professional to learn how to handle data on their own.

Specialization was mentioned as an asset: "when you are more involved in a field usually what you will produce in that field will be more reliable and more complete than a person who is not systematically involved in it", as P14 stated, continuing with the argument that data journalists are journalists that deal with the retrieval of data. To contrast this opinion, P5 contributed to the discussion with the remark that the previously mentioned task could be performed by an analyst and not a journalist. However, again, it was not clear what data journalism is, and the reasoning was that a data journalist will handle someone else's retrieved data and finally will provide the information to the audience under his/her professional view. It was supported that the whole conception is rather a tool that all journalists should possess and not a specific professional category.

Finally, it was claimed that data journalism has always existed, since the point for a journalist is to gather data that he/she will use in the reporting; the only difference between the present and the past is the ease of its acquisition due to the vast amount of information stored on the web.

### 4.2.2. IQ2_2: In What Ways Could Data Journalism Be Helpful for the Public?

Prior to collecting the answers to this question, the moderator asked the participants about their age groups. Most of them (13 out of 14) were between the ages of 20 and 24, therefore belonging to Generation Z. It is commonly discussed that due to the proliferation of content on the web, the audience's attention span is becoming shorter and shorter. In particular, younger people are easily distracted, and they prefer to watch content (videos and images) rather than read long texts (Singh and Dangmei 2016; Podara et al. 2021b).

The visualization of the report was the most common answer among almost all the participants. As P1 stated, data journalism can be helpful "*to better understand an article with data visualization*". Generally, the students' view was that by providing visual components, besides giving a more comprehensible viewpoint, the journalist may also appear more convincing regarding the objectivity of the report. Their point was based on the perception that visualization brings proof of more work on the subject and evidence of the research. As P5 argued, "*if I was looking at an article, I think it would seem more credible if there was a table with some data [. . .] if there is a pie chart or a table, and their reference of course, it would definitely make me more aware, and I would read the whole of it*".

The moderators tried to understand the participants' viewpoints and expectations regarding the overall use of statistics, journalism included. The discussion that followed revealed that, although visualization through statistical charts could be more credible, at the same time, it can be as unreliable as plain text. P4 clearly said that "*if we want to draw the conclusions we want, we can do so with pie charts as well, that's what I believe*". A point that

was made was that the form of visualization should be the correct one according to each case, following journalistic ethics.

A final aspect that was presented was the need to overcome the contemporary short attention span, due to a lack of time, and the ways that data journalism could aid in this direction with visualization. P9 argued that "*maybe a good visualization can keep put our reader, magnetize him/her and let him/her understand in very few words what we want him/her to understand*".

### 4.3. IQ3: Would You, as Young Journalists, Like to Pursue a Career as Data Journalists, and Do You Find It Difficult to Do So?

The next topic was focused on the students' attitudes regarding the possibility of pursuing a career as a data journalist and the probable encountered difficulties. After a first short conversation, the moderators asked the participants to state whether they would like to be involved professionally as data journalists, and only three out of the fourteen replied that they might be interested in it and only one would be very interested.

Their main argument was their impression that data journalism is a rather "raw" and non-creative work area, and this was the reason that they felt indifferent. They did not believe that there was great difficulty in engaging with data journalism and they felt that it was a useful occupation. However, as P4 stated, "*I find it boring […] and its difficulty may come from the fact that it is boring to sort the data and do all this work*". They considered that there was accessibility to data, and they even mentioned open data, but they felt that there were too many software tools that can be used. Despite the availability of online tutorials, they were still not interested in working as data journalists. Of course, they realized that in the event that they were required to do so by duty to aid their careers, they would do so.

As already mentioned, only one participant, P13, was keen on working professionally as a data journalist and stated, "*I don't think that the data and dealing with the data are uncreative and raw. I just think that you can use them in many ways. Personally, I am very interested in this category. So, I would definitely like to learn a lot more because I know almost nothing*".

### 4.4. IQ4: Could the Employment of Data Journalism Cause Problems in Society?

The last part of the discussion, prior to attending the 30-min presentation on data journalism based on Data Journalism—An Introductory Guidebook by the Al Jazeera Media Institute, was centered on the problems that the employment of data journalism could bring to society according to the participants' viewpoints. There was no reference to specific problems, only the problems that journalism could cause, such as exposing people or situations.

Afterwards, a 30-min presentation on data journalism that included many applied examples, as well as specialized tools/software that are employed in this form of journalism, was performed by the moderators. Consequently, the second part of the focus group survey commenced by asking whether the students had changed their viewpoints on data journalism, and ten out of the fourteen replied affirmatively.

### 4.5. FQ1: Did You Learn Something You Didn't Know about Data Journalism and Has Your Viewpoint on Data Journalism Changed? Why?

To the question of whether anything had been added to their knowledge of data journalism, most of the participants agreed that there were many elements that they were not aware of, further commenting that their viewpoint had been clarified and enriched. P1, who, prior to the presentation, characterized data journalism as "raw" and not creative, stated that "*I've decided that data journalism isn't so raw after all. I found it interesting in some parts, maybe it was something I would choose, combined with something else of course, in the future*". Almost all said that data journalism was, after all, a creative form of journalism, as P5 added, "*[…] the way I understand it, the data journalist is something between an analyst and a regular journalist. Meaning he/she gives color to percentages and numbers that someone else might not pay so much attention to, and it can be a whole story based on the data*".

*4.6. FQ2: Would You Like to Pursue a Career in Data Journalism? Why?*

Following the former question, the conversation continued, adding one more factor concerning the possibility of pursuing a career in data journalism after having clarified it. The students admitted that their perception had changed and that, with proper guidance and training, it would be among the goals that they would like to pursue in the future. The moderators fueled the discussion by asking whether the word "data" was the one that discouraged them from being involved in data journalism. P5 admitted that the meaning of the word "data" seems rather "cold"; however, "*Data Journalism is more interactive, it can give more to the story in this way and it is not exactly data, it is the same story simply more enriched with what the technology with the tools gives nowadays*".

Nevertheless, the difficulty of multitasking in the journalistic profession discouraged the participants, since, as P11 pointed out, "*[. . .] journalists have to do a little bit of everything. This obviously adds a level of difficulty, thus making the already difficult role of a journalist even more difficult*". The job complications of data journalists, which exceed those of a normal journalist, caused the students to realize that there were many issues to be considered prior to reaching a decision. One of the issues was that the participants were not sure that, in Greece, they could be members of a broader team that would include other specialties as well (e.g., programmers and designers).

Furthermore, the level of expertise needed was mentioned as a type of difficulty, especially for older journalists, for whom is difficult to adapt to this new generation of content. Four of the participants stated that, in their opinion, the main problem is the demand for technical knowledge.

In the study of Sandoval-Martín and La-Rosa (2018), it was recorded that the profile of a data journalist that possesses many skills is not accurate and that other specialties (programmers and designers) work together with them to present a collaborative project.

*4.7. FQ3: What Do You Consider to Be the Biggest Challenge for a Career in Data Journalism?*

The last question was focused on the challenges of data journalism, and it was smoothly connected with the previous one since the difficulties had already been mentioned. Fantasy and creativity were stated as the greatest challenges since they are both inherent characteristics, as expressed by the students. P12 said, "*[. . .] so I think that the creativity is what is missing, but if we can combine the creativity with the knowledge and the willingness for data journalism, after that it will be like a game*". The moderator tried to intrigue the participants by mentioning the ease of finding new presentation forms nowadays, since there are a number of them all over the Internet. Therefore, creativity may emerge as a variety of images are provided. The students also stated that what they themselves wished to see in information presentation could be a trigger, and this could encourage them to learn more and proceed one step further. However, one of them pointed out that not all topics could be presented via visualizations and graphics, and it is a challenge to find the topics that justify this type of handling.

From another aspect, two students referred to the ability to organize their thoughts and results and, in general, to develop organizational skills to deliver a complete outcome as a challenge. From the same perspective, P4 said, "*I think that the biggest challenge is not to get lost in the numbers and conclusions and to be able to extract valid data that is at the same time interesting*". The appropriate expertise and literacy in this area, along with the willingness to break out of the usual boxes and innovate on a topic that is not covered either by traditional media or social media, is also considered a challenge.

The participants' anxiety about the acceptance and comprehension of their work was also expressed. They stressed that they were not sure whether the volume of the work needed to present a topic as data journalists would be understood and accepted by the public.

P11 condensed the aforementioned notions and said, "*I think that the challenges are enough, but I think that it is multifactorial issue. Meaning it is not only the part of creativity, but also originality, the organization at work, the team spirit too. It is definitely multifactorial*"

*issue*". The last student that replied to this question referred to the financial issue. The point made was whether being involved in data journalism and having acquired all the skills and knowledge would result in a satisfactory profit.

## 5. Discussion

This study aimed at presenting the viewpoints of Journalism and Mass Communication second-year students on data journalism through a dual perspective, comprising their initial considerations on the issue and their thoughts after a learning session. The focus group survey was selected as the most suitable method since it allows participants to freely express themselves, creating a friendly environment that leads to a vibrant discussion. In the context of answering the research questions, the participants willingly replied to the moderators' queries, and, through the discussion and the presentation, new ideas and thoughts emerged for the students, as admitted by them.

Regarding RQ1 (What are the perceived attributes, benefits and challenges of data journalism by journalism students who have no prior experience with the subject?), the participants were asked to contribute their thoughts in a conversation that was started by a set of initial questions that were prepared by the researchers. At the beginning, the students referred to the fragmentary features of data journalism, and one of their original viewpoints was that data journalists belong to a certain category of journalists that are occupied exclusively with collecting data, which are afterwards forwarded to others to report. Uskali and Kuutti (2015), in their study, also found that even professional journalists do not agree on whether data journalism is only for specialists or for everyone.

Furthermore, the participants stated that data in data journalism are derived from empirical research conducted via questionnaire dissemination and that the presentation forms include only statistical charts, such as pie charts. Regarding the features of the term, the recording of the data and its presentation in a visual form were described almost unanimously. The great asset of the focus group survey is the ability to contribute to the conversation with even small, personal aspects. In this way, single words, such as the analysis, assortment and editing of the data, were mentioned, adding further insights to the process.

Data journalism was considered helpful to the public by the students that took part in the research, especially regarding the visual components of the report, which was the most common answer since, besides providing clarity, they allow for a more convincing presentation. Furthermore, visualization can be a solution to the short attention span of the contemporary audience.

Among the most interesting findings was the fact that the participants, upon hearing the word "data", categorized data journalism as "raw" and not creative; thus, only one out of the fourteen participants stated that he/she would like to be involved professionally, and three more claimed that they would consider the possibility. Although they felt that it might be a useful occupation and not difficult to pursue, they categorized it as boring.

Regarding RQ2 (What are the perceived attributes, benefits and challenges of data journalism by journalism students after a learning session on the subject?), their original perspectives on data journalism were significantly enriched and, at some points, altered as, after the presentation, they realized that when all the meaningful patterns in a data structure are combined, they could be evaluated in a creative form. After attending the presentation that encompassed audiovisual modes besides textual ones, they understood that data may be derived from many sources and be presented in many formats.

They recognized that there are many parameters that define data journalism besides merely the word "data", which seemed rather restrictive for the students, who felt more comfortable when presenting their ideas through writing. In agreement with Weiss and Retis (2018), the students in this study, too, considered that data journalism is challenging and connected to difficult mathematical and statistical work, which limits the willingness to learn more about it and deal with it. Professional journalists, too, are hesitant, to a degree, in regard to computational literacy and working with numbers, thus making them

reluctant to be involved with data journalism (Borges-Rey 2020). Therefore, it was not a surprise that the participants of the focus group survey, prior to attending the 30-min presentation, believed that data journalism was rather "raw" and not creative, since it deals with data that are associated with numbers. Afterwards, it was more obvious that it may relate to thought-provoking social dimensions, which was an important issue to them, and not merely with the quantification of parameters. They realized that engaging, new, hybrid forms of stories can be produced in meaningful ways.

Furthermore, it was revealed that journalism projects and storytelling through the means provided by data journalism can be, in many cases, compelling and interactive, matching with the characteristics of the contemporary active news users, who seek information on news websites or platforms and explore data on their own (Karypidou et al. 2019; Podara et al. 2021a, 2021b).

The participants realized the demand for multidisciplinary teamwork in the contemporary newsroom, involving analysts and designers, in the cases of large journalistic projects (Sandoval-Martín and La-Rosa 2018; Veglis et al. 2022). However, they were concerned about whether this would be possible while working in Greek media organizations. From an educational point of view, the assignment of joint essays where each student could contribute with his/her more advanced skills (i.e., data analysis, web development, visualization, audiovisual production, text writing, etc.), all contributing to one output, could allow them to engage in collaboration and realize all the aspects needed (Hewett 2016).

The final question, RQ3 (How was the knowledge on data journalism affected by the learning session and the group discussion?), was oriented towards reaching conclusions regarding the learning procedure via the specific presentation after having already discussed the issue. This approach of effective teaching using audiovisual media communication can be very helpful in delivering knowledge on ambiguous topics, especially when considering the inherent and specific characteristics of the audience (Nicolaou and Kalliris 2020; Nicolaou et al. 2022; Galatsopoulou et al. 2022). It is the authors' belief that the discussion that proceeded the presentation was an important means of gaining knowledge afterwards, since the dissemination of information was based on issues previously discussed and interest in the subject was already aroused, as the students were actively involved. Although the conversation took place in a survey framework, the questions posed elicited students' interpretations in a lively environment, complying with the discussion method in teaching (Abdulbaki et al. 2018). Furthermore, the arrangement of the seating in a circle-like form aided in the teamwork dynamic of revealing personal thoughts within a group. Overall, discussion is a valuable method in the teaching of social studies (Rahman et al. 2011). Pretest questions that detect the extent of students' learning status allow them to play an active role and think broadly; therefore, learning becomes student-centered (Johanna et al. 2023).

Data journalism, as it evolves, may promote the idea of a transparent society as more tools involving both clearer presentation and verification and fact checking can be employed in the procedure of news reporting, increasing journalism's legitimacy among audiences (Lewis and Westlund 2015; Zamith 2019). The participants expressed their viewpoints on the subject, especially with regard to clearer presentation, and argued that more credible articles could be produced using data journalism, since information may be visually displayed. Nevertheless, they emphasized the importance of the ethical aspects of the profession, since, as pointed out, "*lies can be told with charts, too*". In the same context of reporting visualization, the participants considered it a powerful asset in overcoming the issue of the short attention span of the public, mainly due to a lack of time.

To conclude, it is the authors' belief that journalism education curricula should incorporate data journalism courses, pointing out the results, besides the tools' employment, since students are more interested in the impact of the story rather than the means by which it is achieved. If they are provided with compelling motivations, they will be more easily engaged in the process of strengthening their skills. Since many of the courses may be elective for students, short presentations indicating what they include, as well as real

work outcomes, could aid them in realizing the ultimate scope and provide a complete impression of what could be achieved through the perception of the specific knowledge.

There were limitations in the study, one of which was the small number of participants and the fact that they were all derived from the same School of Journalism, prohibiting the generalization of the results. In future work, conducting further research on other university students could enrich the findings and provide a more thorough insight into both the issues of data journalism perceptions and journalism education.

**Author Contributions:** Conceptualization, E.G.; methodology, E.G. and M.M.; validation, E.G. and M.M.; formal analysis, M.M.; investigation, E.G. and M.M.; data curation, E.G. and M.M.; writing—original draft preparation, E.G. and M.M.; writing—review and editing, E.G. and M.M.; supervision, E.G.; project administration, E.G. and M.M. All authors have read and agreed to the published version of the manuscript.

**Funding:** This research received no external funding.

**Institutional Review Board Statement:** Ethical review and approval were waived for this study due to consent provided by the adult participants prior to joining the focus group survey, which has been audio-recorded and is available upon request.

**Informed Consent Statement:** Informed consent was obtained from all subjects involved in the study.

**Data Availability Statement:** Data are available upon request.

**Acknowledgments:** The authors would like to thank students of the School of Journalism and Mass Communications at Aristotle University of Thessaloniki, Greece for their participation in the survey; without their help this work could not be completed.

**Conflicts of Interest:** The authors declare no conflict of interest.

## Note

1 https://diavgeia.gov.gr/ is a website created by the Greek Ministry of Digital Governance that aims to publish on the Internet the decisions of the Government Bodies and the Administration (accessed on 1 November 2023).

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
