# Peer review of "Understanding and Enhancing Journalism Students’ Perception of Data Journalism"

_journalmedia, doi:10.3390/journalmedia4040078_

Round 1
Reviewer 1 Report
Comments and Suggestions for Authors
The article addresses a question of evident relevance and provides substantial findings on a field of study such as data journalism education. The authors present their ideas in a clear and concise manner, effectively conveying their message to their target audience. The paper reflects the authors' dedication and hard work, as it is evident that they conducted thorough and comprehensive field research to ensure that their findings were accurate and reliable.
However, the following recommendations are made:
Introduction
The structure of the article is appropriately presented, however, it lacks the articulation of objectives or research questions to meet the standards of academic research. These statements should serve to guide the results and also be reflected in the discussion.
Literature Review
The bibliography is sufficient and up-to-date. Nevertheless, there is a need to refine the placement of some references to clarify the source of information. For example, at the beginning of this section (lines 52-61), three references are grouped together which should be distributed when the specific information they contain is used.
Just one observation regarding the bibliography: there is an article that delves into the results of the Datajournalism.com survey mentioned.: Simona Bisiani, Andrea Abellan, Félix Arias Robles & José Alberto García-Avilés (2023) The Data Journalism Workforce: Demographics, Skills, Work Practices, and Challenges in the Aftermath of the COVID-19 Pandemic, Journalism Practice, DOI: 10.1080/17512786.2023.2191866
Methodology
The process is clearly described, although it should detail or emphasize more strongly aspects such as the number of students participating in each focus group. The selection of the method is commendable, as it is not very common and can provide more information than more conventional methods such as interviews. However, its major advantage should be emphasized: the promotion of discussion among students. The results of this method should also be analyzed and discussed in the final part of the article.
Reviewer 2 Report
Comments and Suggestions for Authors
The title is very narrow. Even if second year journalism students were analysed, perhaps one can say Journalism students? Is second year, of specific importance? Otherwise just remove this word.
The introduction is very short. While it certainly informative it lacks of relevant references outlining how data journalism has been studied by scholars, how it is taught in journalism schools and what the scientific problem behind this study is. It is also interesting to know more about the Greek context in terms of data journalism, how it is practiced and how it is taught in relation to the chosen scientific problem that the study intends to solve.
The literature is nicely written, but is only focused on documenting the development of data journalism. This review should also include a focus on the development of scientific studies of data journalism, in particular studies that have been influential. For example, a study commissioned by Google News Lab is not a scientific study, neither are the survey by the European Journalism Center. These reports are relevant for motivating the practice, but say nothing about the scholarly community that have studied data journalism for over a decade. Most of the relevant work on data journalism that has been published is missing from this study.
I appreciate the section on data journalism education very much. I think the right studies have been cited here. Please note however, that the studies quoted here, are not the leading studies in the scholarly community on data journalism.
The method section is well written, but a study based on one focus group is not enough. This study needs more data. Three focus groups should be the minimum.
Round 2
Reviewer 2 Report
Comments and Suggestions for Authors
I want to congratulate the authors on this revised version. My only comments is that scraping was misspelled on row 42 on page 1.